# A Multidisciplinary Investigation into the Talent Development Processes in an English Premiership Rugby Union Academy: A Preliminary Study through an Ecological Lens

**DOI:** 10.3390/sports10020013

**Published:** 2022-01-18

**Authors:** Francesco Dimundo, Matthew Cole, Richard C. Blagrove, Kevin Till, Adam L. Kelly

**Affiliations:** 1Research Centre for Life and Sport Sciences (CLaSS), Department of Sport and Exercise, School of Health Sciences, Birmingham City University, Birmingham B15 3TN, UK; matthew.cole@bcu.ac.uk (M.C.); Adam.Kelly@bcu.ac.uk (A.L.K.); 2Worcester Warriors Rugby Football Club, Sixways Stadium, Worcester WR3 8ZE, UK; 3School of Sport, Exercise, and Health Sciences, Loughborough University, Loughborough LE11 3TU, UK; r.c.blagrove@lboro.ac.uk; 4Carnegie School of Sport, Leeds Beckett University, Leeds LS1 3HE, UK; K.Till@leedsbeckett.ac.uk

**Keywords:** psychology, socioeconomic, social identity, physical, cognitive skills

## Abstract

(1) Background: The progression of youth rugby union (RU) players towards senior professional levels can be the result of various different constraints. The aim of this study was to examine characteristics that differentiated playing positions and player rankings in an English Premiership RU academy. (2) Methods: Thirty players (mean age = 18.5 ± 2.8 years) were divided by playing positions (forwards = 18, backs = 12) and ranked (one to thirty) by coaches based on their potential to achieve senior professional status. Players were analysed across 32 characteristics from eight overreaching factors based on task, environmental, and performer constraints. MANOVA and ANOVA were used to calculate differences among variables in players’ positions (i.e., forwards vs. backs) and ranks (i.e., top 10 vs. bottom 10), with a Welch’s *t*-test applied to identify individual differences amongst groups and effect sizes calculated. (3) Results: Large effect sizes were found between groups for socioeconomic, sport activity, anthropometric, physical, and psychological factors. Moreover, environmental and performer constraints differentiated playing positions, whereas task and environmental constraints discriminated player ranks. (4) Conclusion: Present findings showed that playing positions and player ranks can be distinguished according to specific constraints.

## 1. Introduction

One of the main challenges of youth development in rugby union (RU) is to predict future talents at both professional club and national governing body levels [1]. The talent development (TD) processes have been observed in sports’ literature [2,3,4,5,6,7] and more recently applied to the context of RU [8]. There is currently an acceptance by clubs and organisations that the progression of RU players towards the top levels of competition is multifactorial, which can be underscored using an ecological dynamics theoretical approach [9]. This implies that developing expertise in RU cannot be the result of a single independent factor and is instead the result of a combination of *task* (i.e., participation history), *performer* (i.e., psychological, technical-tactical, anthropometric, physiological), and *environmental* (i.e., relative age, sociocultural) constraints [10,11].

The weaknesses of the TD processes in RU are represented by the limited multidimensional examinations of players within the current literature [8,12,13,14], which often do not take into account all the ecological constraints or analyse players’ positions or playing levels [12]. This is despite evidence indicating that athletes’ progression is largely affected by a range of factors, such as anthropometric [15], physiological [16,17], psychological [18,19], social identity [20], socioeconomic [21,22], and perceptual-cognitive expertise (PCE) [1,23] traits. Moreover, it has been found that the participation in adult-led practice and peer-led play in sport-specific and multisport activities [24,25], as well as the accumulation of hours of game-exposure at different ages [26], can impact the development of a young player. In addition, population density in the town of growth [27,28] and levels of deprivation [29] have been shown to have a significant impact on the TD opportunities and outcomes in RU. Indeed, researchers have recommended that future investigations in RU should consider these aspects in unison when studying professional academy contexts in order to better understand the holistic demands of the TD process [3,30].

A RU team is comprised of 15 playing athletes with a maximum of eight substitutes. The squad is generally split into forward (e.g., props, hooker, flankers) and back (e.g., inside centre, outside centre, full back) players. Forwards are those who normally engage first with opponents and are involved in set pieces and close high-force collisions. In contrast, backs are players who tackle the opposition at a later stage of the game, are engaged in rapid actions, and cover longer distances in high speed running [31,32]. Since the two main playing positions require the development of specific characteristics, a different TD path is often needed for these players [31]. Existing investigations on long-term athlete development (LTAD) pathways in RU have yet to elucidate the most suitable qualities to train forward and back players through an ecological lens [33]. In addition, there are only limited suggestions on how to differentiate playing levels based on player rankings in RU [34]. Indeed, questions remain surrounding the most appropriate processes that facilitate players’ progression towards RU senior professional status, since sport organisations’ *modus operandi* can often result in missing future professionals due to the pyramidal structure of the talent identification (TID) system, in which, at each stage of selection, the number of places for players to follow a development path decreases [35]. Unfortunately, existing research is yet to report a multidisciplinary investigation based on the aforementioned areas that are important for TD in RU. Thus, the present investigation aimed to examine a range of task, environmental, and performer constraints in an English Premiership RU academy. Specifically, both playing positions (i.e., forwards and backs) and player rankings (i.e., top-10 potentials vs. bottom-10 potentials) were analysed to: (a) offer a preliminary study to better understand the TD processes in RU, (b) provide professional RU academies a novel approach of assessing players, and (c) establish a methodological framework that may be useful for other researchers in the future.

## 2. Materials and Methods

### 2.1. Participants

Thirty players (under 16 (U16) = 11, U18 = 9, U21 = 10) from an English Premiership RU academy agreed to participate in this study. Table 1 reports the descriptive statistics of the participants. All participants were analysed based on playing position to compare possible differences (forwards = 18, backs = 12). They were also ranked on their potential to become a senior professional RU player, regardless of playing position and age, from one to thirty by three Level 4 academy coaches. Coaches ranked players using subjective criteria based on both their own vision of the game and personal philosophy of coaching. This produced a linear classification of higher-ranked players down to their lower-ranked peers, who were then split into thirds using tertiles. This created a cohort of ‘top-10 potentials’ (*n* = 10), who represent the top third, and a cohort of ‘bottom-10 potentials’ (*n* = 10), who represent the bottom third. This enabled a distinct comparison between the higher- and lower-ranked potentials across the group, with the middle third discarded from the player rank analysis (*n* = 10). Ethical approval was granted by the Faculty of Health, Education, and Life Sciences Research Ethics Committee at Birmingham City University.

### 2.2. Procedure

Data were collected during the first 9 weeks of the 2019 pre-season where athletes were tested before afternoon training. Participants were instructed to follow a standardised training and recovery procedure in the 48 h before each physical testing session. All physical tests were proceeded by a familiarisation trial and were conducted during the same day. Each anthropometric and physical test was explained and demonstrated with physical assessment preceded by a standardised RAMP warm up, a type of activation similar to what players usually perform before training and competition (e.g., mobility, dynamic stretching, low level plyometrics, and running drills). The PCE video simulation test was performed in a room that comprised a setting similar to a classroom to enhance players’ concentration and comfort at the club. Psychological, socioeconomic, social identity, and participation history were collected using validated questionnaires distributed via an online platform (Online surveys Jisc, Bristol, UK), which participants were asked to complete in their own time. In total, players were analysed over 32 characteristics from eight overreaching factors based on task (i.e., participation history and sport activities), environmental (i.e., socioeconomic), and performer (i.e., anthropometrical, physical, PCE, and social identity) constraints.

### 2.3. Task Constraints

#### Participation History

An adapted participation history questionnaire was used to gather the participants’ engagement in activities throughout their youth [36]. Following the Developmental Model of Sport Participation (DMSP [24,37]), data were collected using estimated time (in hours) spent in RU competition, coach-led practice, and peer-led play between the ages of 8–11 and 12–15 years. The number of sports played until the age of 15 years was also recorded to provide information on the variety of players’ motor ability and competency in basic and complex motor athletic skills. This study followed guidelines indicated previously [36].

### 2.4. Environmental Constraints

#### Socioeconomic Factors

The town where participants spent the most of their life during childhood and adolescence was recorded via an online questionnaire. The number of inhabitants and index of multiple deprivation decile was calculated using the UK government data available online [38]. The size of the town was ranked using the classification adopted by Cobley et al. [27], where the crescent number of inhabitant per town was labelled according to a number ranging from 1 to 5: 1 = 0–9999, 2 = 10,000–19,999, 3 = 20,000–49,999, 4 = 50,000–99,999, and 5 = 100,000–199,999. Moreover, according to the government norms, the index of multiple deprivation (IMD) decile reflected the players’ socioeconomic situation from the most deprived (scored with ‘1’) to the least deprived (scored with ‘10’).

### 2.5. Performer Constraints

#### 2.5.1. Anthropometric

Body mass and height were measured to the nearest 0.1 kg and 0.1 cm using calibrated Seca Alpha (model 220) scales and Seca Alpha stadiometer (Seca, Hamburg, Germany), respectively.

#### 2.5.2. Physical

Isometric hip extension (IHE) strength was measured using a portable Takei Back and Leg Dynamometer (Takei Scientific Instruments Co., Ltd., Tokyo, Japan), whereby participants stood on a portable platform with knees fully extended, back in a neutral position, and hips flexed. The length of the handle’s chain was set according to the participant’s height by asking the subject to stand with extended knees. The handle was then positioned at the height of the intra-articular space of the knee joint. Subjects were instructed to lift vertically in order to generate isometric contractions of the extensors of the knees, hips, and lower back while pulling the handle as hard and as fast as possible for 5 s. Dominant handgrip strength was measured using the Takei 5401 Handgrip Dynamometer (Takei Scientific Instruments Co., Ltd., Tokyo, Japan). Participants performed the test sitting and holding the shoulder at 0° flexion, abduction and rotation, the elbow flexed at 90° and wrist positioned between 0° and 30° dorsiflexion and between 0° and 15° of ulnar deviation. Participants were instructed to ‘squeeze’ as hard as possible for 5 s and the best results of three attempts was recorded, with a 3-min rest between tests. Thus, the muscle strength primarily generated by the flexor muscles of the hand and the forearm could have been recorded. Strong verbal encouragement was provided during each repetition. These tests followed standardised validated procedure explained in previous literature [39,40]. All participants’ positions for both the isometric hip extension and handgrip strength test were checked in a previous recording. The dominant hand was determined by asking the participants which hand they normally write with. Recorded measure from the two dynamometers consisted of the maximal force expressed in kg.

All participants were familiar with the CMJ as this was used frequently in testing and training at the club. Players performed three trials of a CMJ by jumping as high as possible while positioned between two parallel infrared beams (Microgate, OptoGait, Italy) and following a standard procedure already used in literature (e.g., [41]). After circa ninety seconds of recovery, players then completed three attempts for the reactive strength index (RSI) test whereby they performed ten consecutive jumps trying to reach maximal height for every bounce whilst spending as little time in contact with the ground between jumps as possible. RSI was calculated for each jump as the ratio between height (in metres) and contact time (in seconds). The best score of the three attempts on both tests was recorded. Peak power was calculated using Sayers equation [42]:Peak power (W) = (60.7∙H) + (45.3∙W) − 2055
where ‘H’ refers to the CMJ height in cm; ‘W’ to body mass in kg.

Relative peak power (W∙kg^−1^) was calculated by dividing peak power by player’s body mass.

Sprint time over 20 m was recorded using timing gates (Brower Timing Systems, IR Emit. Draper, UT, USA). Timing gates were placed at the starting point and at 20-m distance. Following the warm-up, participants completed three maximal sprints from a staggered start with a 3 min passive rest between attempts. Each sprint started behind the initial timing gate (0.3 m), with players instructed to set off in their own time and run maximally through the final 20 m timing gate. Participants’ starting point was checked before they were allowed to proceed. The best of the three attempts was taken for analysis with times measured to the nearest 0.01 s. Momentum was simply calculated by multiplying body mass and estimated final velocity over 20 m sprint, as previously used [16].

The 30–15IFT consisted of a 30 s shuttle run over a 40 m distance, interspersed with a 15-s recovery. The test began at 8 km·h^−1^ and increased by 0.5 km·h^−1^ at each successive running shuttle. All procedures were followed as reported in previous literature [43]. The test was terminated when participants were no longer able to maintain the imposed speed of the test or when they did not reach a 3 m tolerance zone on three consecutive occasions. The velocity from the last completed stage was noted and used to the estimate *V*O_2_max (mL∙kg^−1^·min^−1^) through the following formula [43]:*V*O_2_max (mL∙kg^−1^·min^−1^) = 28.3 − (2.15∙G) − (0.74∙A) − (0.0357∙W) + (0.0586∙A∙V_IFT_) + (1.03∙V_IFT_)
where ‘V_IFT_’ is the final running velocity; ‘G’ refers to gender (male = 1; female = 2); ‘A’ is age; ‘W’ is subject’s body mass (kg).

#### 2.5.3. Perceptual-Cognitive Expertise (PCE)

A perceptual-cognitive video simulation test was used to examine the participants’ decision-making skills based on a combination of tactical situations, which have been used in RU literature [1] and demonstrated to produce valid and reliable measures for PCE research in several sport environments, e.g., [44]. Game situations of fifteen video clips were chosen from live rugby match footage, filmed from different elevated angles to provide a wide-range view of the pitch. Following moments of build-up play, the screen unexpectedly froze for 8 s prior to a critical decision-making moment. A game-related question with four possible options appeared and participants were required to select an answer on their response sheet before the next clip automatically began. As per examination conditions, participants were seated and were unable to engage with each other. Participants overall score was ranked using percentiles (i.e., 90th, 75th, 50th, and 25th) and then classified (i.e., 1 = excellent, 2 = good, 3 = average, 4 = low, and 5 = poor) for analysis. The total accuracy of the participants’ responses was recorded for analysis.

#### 2.5.4. Psychological Characteristics of Developing Excellence Questionnaire Version 2 (PCDEQ2)

To measure psychological characteristics, the seven factor (factor 1 = adverse response to failure, factor 2 = imagery and active preparation, factor 3 = self-directed control and management, factor 4 = self-directed control and management, factor 5 = seeking and using social support, factor 6 = active coping, and factor 7 = clinical indicators) and 88 item PCDEQ2 was used [19]. The answers were ranked with a Likert score ranging from 1 (‘very unlike me’) to 6 (‘very like me’) and then were converted into final scores on the seven factors. This conversion finally led to a score of one to ten for each of the seven items, as explained by Hill et al. [19].

#### 2.5.5. Social Identity Questionnaire for Sport (SIQS)

The SIQS was used to evaluate players’ social identity within their respective academy team. Nine items in a Likert score system (1 = ‘strongly disagree’ and 7 ‘strongly agree’) reflected three underlying dimensions: (a) in-group ties (items 1–3), (b) cognitive centrality (items 4–6), and (c) in-group affect (items 7–9). SIQS total score was also calculated [20]. These data were collected via an online questionnaire that players were requested to complete in their own time.

## 3. Statistical Analysis

The Shapiro–Wilk test was used to check data normal distribution. Anthropometrical, physical, psychological, PCE, and SIQS scores were then normalised using *z*-scores (*z* = (*x* − *μ*)/*δ*), where *x* is the raw score, *μ* is the population (U16, U18, and U21) mean, and *δ* is the population standard deviation. A multivariate analysis of variance (MANOVA) was used to calculate the difference among the combined participation history, socioeconomic, social identity, psychological, anthropometric, and physical factors between both forwards and backs and top-10 and bottom-10 potential players. A one-way analysis of variance (ANOVA) was used to explore the differences for the cognitive test as it was comprised of one variable. A Welch’s *t*-test was then conducted for all the variables to compare differences among players’ positions and ranks. A Cohen’s *d* was also used to calculate the effect size of these factors. Cohen’s *d* effect size was calculated as reported in previous literature [45] with threshold values of 0.20 (small), 0.50 (medium), and 0.80 (large), with corresponding 95% confidence intervals (CIs). Significance was set for an *α* level of 0.05 with the statistical analysis conducted using IBM SPSS Statistics Version 24 (SPSS Inc., Chicago, IL, USA).

## 4. Results

The descriptive statistics are reported in Table 1. The MANOVA for training type factors, socioeconomic factors, social identity factor, psychological factors, anthropometric and physical factors, and the ANOVA for cognitive factors are reported in Table 2. The Welch’s *t*-test analysis is reported in Table 3.

## 5. Forwards vs. Backs

Results showed that there was a significant difference between playing positions for both anthropometric (*p* < 0.001) and physical (*p* = 0.004) factors. The Welch’s *t*-tests reported large differences between players for IMD decile (forwards = 7.7 ± 1.8 vs. backs = 6.3 ± 1.2; *p* = 0.020, *d* = 0.88), body mass (forwards = 98.7 ± 11.6 kg vs. backs = 85.4 ± 7.5 kg; *p* < 0.001, *d* = 0.82), IHE (forwards = 144.1 ± 16.7 kg vs. backs = 131.5 ± 20.2 kg; *p* = 0.013, *d* = 0.81), CMJ (forwards = 35.6 ± 5.7 cm vs. backs = 41.3 ± 3.5 cm; *p* = 0.050, *d* = 0.73), peak power (forward = 4585 ± 654.9 W vs. backs = 4323 ± 476.0 W; *p* < 0.001, *d* = 2.00), relative peak power (forwards = 46.4 ± 3.6 W/kg, backs = 50.6 ± 2.3 W/kg; *p* = 0.041), 20 m momentum (forwards = 635.4 ± 76.7 m·s^−1^ vs. backs = 574.6 ± 57.2 m·s^−1^; *p* < 0.001, *d* = 0.89), *V*O_2_max (forwards = 47.6 ± 5.0 mL∙kg^−1^·min^−1^ vs. backs = 52.7 ± 3.1 mL∙kg^−1^·min^−1^; *p* = 0.012, *d* = 0.98 ), and factor 6 (forwards = 4.4 ± 0.5 vs. backs = 4.7 ± 0.6; *p* = 0.043, *d* = 0.80). In addition, there was no significant differences between positions for the other variables.

## 6. Top 10 vs. Bottom 10

When examining groups based on coaches’ rank, the analysis displayed statistical significance for socioeconomic (*p* = 0.049) and sport activities (*p* = 0.018) cumulative variables. The Welch’s *t*-test showed significant differences among four different factors, whereby the top-10 players: (a) came from a more deprived area (6.7 ± 1.5 vs. 8.3 ± 1.2; *p* = 0.015, *d* = 0.79), (b) were more exposed to hours of rugby game when they were between 8 and 11 years old (120.7 ± 52.3 vs. 59.8 ± 24.3 h; *p* = 0.003, *d* = 0.80), (c) accumulated greater amount of time in training led by peers between ages 12 and 15 years (311.5 ± 274.8 vs. 124.1 ± 48.3 h; *p* = 0.038, *d* = 0.97), and (d) were faster over 20 m sprint (2.97 ± 0.09 s vs. 3.18 ± 0.19 s; *p* = 0.049, *d* = 0.83) compared to bottom-10 players. Moreover, despite small to moderate effect sizes among other variables, these were not statistically significant.

## 7. Discussion

Key findings revealed that environmental and performer constraints differentiated players based on positions. Academy forwards came from less deprived areas, were heavier, stronger, more powerful, and possessed greater momentum. Backs possessed greater relative peak power, RSI, *V*O_2_max, and were characterised by superior active coping strategies (PCDEQ2 Factor 6) compared to forwards. Moreover, task and environmental constraints discriminated player ranks, whereby the top-10 potential players came from more deprived areas, were exposed to more RU competition between ages 8 to 11 years, accumulated a greater amount of engagement in peer-led play between ages 12 and 15 years, and were significantly faster over the 20 m sprint when compared to the bottom-10 potential players.

The IMD decile indicated that forwards originate from less deprived areas compared to backs (i.e., higher IMD score), possibly implying developmental differences in these players. Previous research from Winn et al. [29] found that more deprived young Welsh players engaged in less sports and accumulated less hours of rugby-specific training. In contrast to Winn et al. [29], however, although the present study revealed that backs originated from a *more deprived* areas, it does not reflect the fact that backs were excluded from sports (mean number of sports = 3.7 ± 1.9) and RU activities (e.g., games, coach-led practice, and peer-led play from U8 to U15), nor were *critically deprived* (e.g., IMD below 5). Several studies have attempted to analyse the influence of socioeconomic status on anthropometrical qualities in young RU players [21,22,46,47]. These investigations revealed that players with a lower socioeconomic status were physically smaller and lighter than those players from a higher status. According to present findings and the importance that some qualities have in characterising players in RU [31], the results on IMD decile provide an important indicator to consider when researching and developing young RU players in relation to their position, suggesting more investigation is needed on this aspect.

When analysing players according to their ranking, top-10 potentials came from *more deprived* areas compared to bottom-10 potentials (IMD decile = 6.7 ± 1.5 vs. 8.3 ± 1.2). Thus, it could be suspected that deprivation may help somehow in shaping characteristics useful for unlocking players’ potential. As explained in the rocky road theory of Collins et al. [48], it is possible that the top-10 potentials had both the opportunity to challenge themselves and to have adequate social support to interpret adversities as positive growth experiences. Moreover, it could be speculated that deprivation reduces the engagement of young players with organised sport environments [29], whereas from another perspective it might increase vital opportunities of practice sport-related activities in deliberate play settings with parents, peers, and siblings [49]. In fact, a more enjoyable and peer-led environment has already been adopted from international professional RU teams to stimulate self-awareness, decision-making, tactical awareness, and in general, athlete functionality in adult players [50]. Therefore, this social discrepancy can lead to the possible theory that the IMD decile variable could help in forming attributes relevant to diverse playing positions (e.g., anthropometric, physical, psychological, social identity, PCE), as well as a higher ranking in RU academies. In this light, professional RU environments could add this parameter in a novel format of players’ assessment.

From an anthropometric perspective, this investigation revealed that forwards were heavier than backs (98.7 ± 11.6 kg vs. 85.4 ± 7.5 kg, *p* < 0.001). This is in agreement with previous results across RU academies [31], senior squads [51], and clubs from different countries [12]. Due to players’ positional requirements, a higher body mass in forwards aids in attenuating impacts during tackles and collisions [52]. The variation in anthropometric measures among playing positions consolidates how forwards and backs require diverse anthropometric characteristics in order to perform position-specific tasks during games [31]. From a ranking viewpoint, although not statistically significant, top-10 potentials were heavier than bottom-10 potentials, indicating this may be important for players to succeed in an academy. Recent studies demonstrated how body mass was pivotal to distinguish selected and non-selected academy players in England [53], predict players’ progression in an Italian academy [54], as well as to discriminate positions in South African [52], Zimbabwean [55], and Argentinian [56] academy environments. Therefore, coaches should consider the importance of body mass in developing players and their progression across an academy. However, practitioners should be aware that players of the same chronological age can differ in their maturity status, and therefore caution should be placed when selecting players based on morphology parameters only.

Physical parameters have been shown to differentiate both playing positions [31] and age-grade players [56], as well as to distinguish levels [53] in RU academies. In the present investigation, forwards were significantly stronger than backs in the IHE test (144.1 ± 16.7 kg vs. 131.5 ± 20.2 kg, *p* = 0.013), demonstrating the importance of this physical characteristic for this playing position. One of the reasons why forwards are typically stronger than backs is because these players are required to produce higher maximal isometric force during games in holding scrums and competing for the ball in rucks and mauls when compared to backs [32,52]. Together, these findings indicate that different aspects of strength should be developed in RU academies according to players’ individual needs.

Sprint momentum has been defined as a key parameter for performance in RU, as well as differentiating playing levels [56,57] and playing positions [31] in various academy settings across the globe. In the current study, forwards performed 20 m sprint momentum similar to results from U18 forwards in a previous investigation [33] (637.6 ± 91.9 m·s^−1^ vs. 635.4 ± 76.7 m·s^−1^). Present results suggest that forwards outperformed backs due to their heavier body mass. Specifically, when a heavier body reaches a higher velocity, it possesses a greater kinetic energy compared to a lighter body. For instance, maximising sprint momentum through increasing body mass while maintaining linear speed capabilities appears to be an important characteristic for forwards to possess, since such a position involves ball carrying in situations where contact is unavoidable [57]. From a ranking point of view, momentum did not statistically differentiate top-10 from bottom-10 potentials, however, top-10 potentials recorded a medium effect size difference compared to bottom-10 potentials, suggesting that this parameter should be trained in TD environments.

Findings from the CMJ and power-related measures reported that backs jumped significantly higher and possessed greater relative peak power than forwards, indicating that these players could had superior jumping technique and were able to express more power per kg of body mass when compared to forwards (CMJ = 35.6 ± 5.7 cm vs. 41.3 ± 3.5 cm, *p* = 0.05, *d* = −0.73; relative peak power = 46.44 ± 3.62 W/kg vs. 50.55 ± 2.27 W/kg, *p* = 0.041, *d* = −0.80). Similar results were found between positions in a LTAD study within RU academies on CMJ [58] and relative peak power analysis [33]. An explanation for backs’ possessing greater jumping performance and relative peak power is that these factors contribute to optimising linear sprints, changes of direction, agility, and to achieving higher speed from different starting positions during games. Similar to the findings of Howard et al. [59], peak power was significantly greater in forwards than backs in this current study (4585 ± 654 W vs. 4323 ± 476 W), indicating that, in general, players from this playing position often rely on this physical parameter during powerful actions of a match (e.g., closer stance explosive tackles). From a rank perspective, although top-10 potentials recorded superior CMJ, peak power, and relative peak power compared to bottom-10 potentials players, it was not statistically significant. Together, these results indicate that the evaluation and development of power-related qualities should be included in the RU TD process.

In the present study, 20 m sprint was the only physical factor that distinguished the top-10 and bottom-10 potentials, whilst no significant differences were found among positions. Sprint time has recently been shown to be a key factor in TID and TD processes in RU [53]. Moreover, sprint ability was linked both to different levels of RU [17,57], as well as different age groups and positions in different countries [31,56,57,60]. Speed has been increasingly recognised as important by RU practitioners since RU games are becoming more dynamic and faster than previous years [61]. Another possible explanation is that, as per body mass characteristics, sprinting speed has been correlated to momentum, which is a key component in RU matches [57]. Therefore, practitioners are encouraged to focus on maximising the development of the different phases of sprint mechanics in academies.

Aerobic capacity was estimated using the 30–15IFT. The only statistically significant difference was found between positions, whereby backs had a greater *V*O_2_max when compared to forwards (52.7 ± 3.1 mL∙kg^−1^·min^−1^ vs. 47.6 ± 5.0 mL∙kg^−1^·min^−1^, *p* = 0.012), which aligns with previous literature [62]. Indeed, backs are generally leaner and have a lower body fat percentage when compared to forwards, which may have facilitated a superior aerobic profile when expressed relative to body mass [31]. Moreover, the specific demands of forwards requires them to cover less distance in a game when compared to backs [63], which may be explained with the present findings. Although not statistically significant, the top-10 potentials possessed greater *V*O_2_max when compared to the bottom-10 potentials (*d* = 0.69), suggesting that this may have a certain degree of importance to differentiate ranks in players. Therefore, aerobic capacity should be trained based on position during a LTAD pathway [35] and be part of an assessment battery in RU.

Previous studies attempted to distinguish psychological traits in different playing positions [34,64,65], ranking [34,64,66], and based on coaches’ perspectives [67] across RU players. Specifically, existing literature shows that forwards generally possess greater psychological skills, such as relaxation, stress reaction, and fear control [64,65] when compared to backs. Indeed, only one study [34] has shown that both forwards and backs possess equally good psychological traits (i.e., determination, goal directedness, self-confidence, concentration, and mental preparation). On the contrary, however, the results from the present study showed how backs were characterised by superior perceived active coping strategies (PCDEQ2 Factor 6) when compared to forwards. It is plausible to suggest that backs may experience more pressurised situations during competitive match-play when compared to forwards, since their role includes critical moments, such as executing penalty kicks and kicking conversions which require quick decision-making skills. Moreover, since the current study showed how backs come from higher deprivation, it could be speculated that a greater perceived active coping was a result of an adaptation to a more challenging socioeconomic environment during their development. However, further research is required to substantiate these suggestions and explore the association between socioeconomic status and the development of psychological characteristics in talent pathways. No significant differences were reported in psychological variables between top-10 and bottom-10 potential players. Thus, the present findings could be used to help explain the role of the environment and psychological development in RU players and guide future research.

With regards to the engagement in sport activities (i.e., game exposure, coach-led practice, and peer-led play), there were no positional differences at both aged 8–11 and 12–15 years. In comparison, however, the top-10 potentials engaged in more hours of game exposure at a younger age (i.e., aged 8–11 years) and accumulated more time in peer-led play during late childhood and early adolescence (i.e., aged 12–15 years) when compared to the bottom-10 potentials. An early exposure to competition has been considered an important part of the athlete development process [26,68], which aligns with the understanding that young players should be exposed to various enjoyable games that gradually produce more demanding performance-specific situations with an older age [18]. Similar to the present results, in a recent meta-analysis from Güllich et al. [68] it was reported that although world champions started their main sport at a later stage in life, higher performing athletes accumulated significant early exposure of their main sport than lower performers (*p* = 0.010; *d* = 0.20). In handball, for instance, Bjørndal et al. [26] stated that an early exposure to the competitive experience represented a vital part for player development towards their high performer status. Thus, coaches should take into account the potential long-term benefits that high-quality game exposure could have on players’ status. With regards to player rankings, the top-10 potentials accumulated a greater number of hours in peer-led play between aged 12–15 years when compared to the bottom-10 potentials. Although these findings report controversies with conclusions of a recent study on athletes’ progression [69], they align with rugby league research that has shown the importance of peer-led activities in development of professional players [64,70]. Thus, more varied learning experiences during early-adolescence could facilitate a later rugby-specific skill learning and refinement [71]. From an ecological dynamic perspective [9,10], it is possible to explain present results through the variation in learning tasks and environments, which may facilitate a players’ ability to adapt their actions in learning and to familiarise their movement across various unpredictable environments (i.e., enhanced athlete functionality, see Rothwell et al., [50]). As such, a players’ later exposure to peer-led play may continue during the transition from childhood to adolescence, which is a crucial stage for young RU players since they are generally selected to be part of a professional academy for the first time (i.e., at U15).

Overall, these findings offer a preliminary study to better understand the TD processes in RU, provide professional RU academies a novel approach of assessing players, and establish a methodological framework that may be useful for other researchers in the future.

## 8. Limitations and Future Directions

One limitation of this study was the small number of participants. A larger sample may have altered the outcomes of the current findings, especially those in relation to ranked players [72]. Another limitation of this study was that no age-related differences were investigated (i.e., it could be possible that different ages influenced players’ ranks). However, the novelty of this study also compares those who have already been selected into an academy environment through analysing potential to achieve senior professional status, rather than the traditional approach of comparing ‘elite’ vs. ‘non-elite’ or ‘selected’ vs. ‘non-selected’, thus further limiting the prospective pool of participants. Moreover, it is important to mention that present results only reflect the status of a single Premiership RU academy, and thus it is possible that this is not representative of other environments in RU. However, other studies surrounding TD in RU [16] and football [44] adopted similar methodological procedures when analysing academies of professional clubs. Furthermore, some data were collected retrospectively (e.g., game exposure, peer-led play, and coach-led training), and therefore recall bias may have influenced findings. Nevertheless, previous research has applied these tools and demonstrated a good level of reliability and validity (e.g., [69]). In addition, due to the large number of data collection methods required to be completed in order to be included in the current study, only those academy players who conducted all the measures were analysed. Therefore, it is important to recognise that this study may have not considered participants whose results may have changed the outcomes should they have completed all the protocols. However, due to these limitations, this study was denoted as a preliminary investigation to ensure the reader acknowledges the exploratory nature of the research being performed. Thus, the present investigation can be used to guide future research methodologies which are encouraged to maintain a multidisciplinary approach and use a longitudinal protocol with a greater and more diverse sample.

## 9. Conclusions

To the authors’ knowledge, this is the first multidisciplinary study that has analysed 32 characteristics from eight overarching factors in an English Premiership RU academy through an ecological dynamic lens. Present findings showed how playing positions can be differentiated by environmental and performer constraints. Moreover, top-10 potential players were distinguished from bottom-10 potential players in task and environmental constraints. Rugby practitioners are encouraged to follow a similar multidisciplinary approach and use these findings as framework when assessing professional academy players. Researchers could also use the methodology employed in this investigation as the basis for future work in this area.

## Figures and Tables

**Table 1 sports-10-00013-t001:** Descriptive statistics for forwards, backs, top-10 potentials, and bottom-10 potentials.

Factors	All Forwards (*n* = 18)	All Backs (*n* = 12)	Top-10 Potentials	Bottom-10 Potentials
	Mean ± SD (*z*-Score)	Mean ± SD (*z*-Score)	Mean ± SD (*z*-Score)	Mean ± SD (*z*-Score)
Age (year)	18.1 ± 3.1	18.4 ± 2.9	19.0 ± 2.9	18.7 ± 2.3
BQs	1.9 ± 1.1	2.0 ± 1.1	1.7 ± 1.0	2.0 ± 1.0
**Task constraints**				
*Participation history*				
Number of sports	2.9 ±1.8	3.7 ± 1.9	3.5 ± 2.1	3.5 ± 2.2
*Sport activities*				
Game exposure U8-U11 (h)	74.1 ± 47.5	99.0 ± 50.1	120.7 ± 52.3	59.8 ± 24.3
Coach-led U8-U11 (h)	300.8 ± 182.3	216.5 ± 131.3	296.4 ± 112.1	216.0 ± 193.8
Peer-led U8-U11 (h)	126.8 ± 159.0	81.0 ± 72.3	139.0 ± 209.1	82.7 ± 62.9
Game exposure U12-U15 (h)	226.1 ± 114.4	222.4 ± 93.0	234.8 ± 122.2	215.5 ± 71.7
Coach-led U12-U15 (h)	411.9 ± 274.1	343.6 ± 150.7	391.0 ± 175.5	368.4 ± 225.3
Peer-led U12-U15 (h)	255.2 ± 233.1	287.5 ± 316.2	311.5 ± 274.8	124.1 ± 48.3
**Environmental constraints**				
*Socioeconomic*				
Town population (AU)	4.7 ± 0.5	4.2 ± 1.0	4.3 ± 1.0	4.3 ± 0.8
IMD decile	7.7 ± 1.8	6.3 ± 1.2	6.7 ± 1.5	8.3 ± 1.2
**Performer constraints**				
*Anthropometric*				
Body mass (kg)	98.7 ± 11.6 (0.606 ± 0.745)	85.4 ± 7.5 (−0.908 ± 0.390)	96.0 ± 11.2 (−0.118 ± 0.906)	94.4 ± 11.9 (0.234 ± 1.017)
Height (cm)	180.4 ± 4.7 (0.025 ± 0.737)	171.9 ± 42.9 (−0.022 ± 1.114)	178.3 ± 6.2 (−0.304 ± 1.057)	163.6 ± 56.8 (0.001 ± 1.017)
*Physical factors*				
Hand grip (kg)	48.2 ± 5.7 (0.022 ± 0.966)	50.4 ± 5.0 (−0.025 ± 1.004)	52.6 ± 4.3 (0.072 ± 1.016)	46.1 ± 5.5 (−0.252 ± 1.102)
IHE (kg)	144.1 ± 16.7 (0.389 ± 0.650)	131.5 ± 20.2 (−0.583 ± 1.091)	147.2 ± 22.8 (−0.088 ± 1.035)	130.5 ± 13.5 (0.057 ± 1.067)
CMJ (cm)	35.6 ± 5.7 (−0.261 ± 0.862)	41.3 ± 3.5 (0.417 ± 1.014)	40.3 ± 4.8 (−0.225 ± 0.999)	35.4 ± 5.4 (−0.209 ± 0.734)
Peak power (W)	4585.53 ± 654.94 (0.539 ± 0.769)	4323.81 ± 476.07 (−0.927 ± 0.695)	4743.9 ± 644.5 (−0.218 ± 1.012)	4339.53 ± 586.3 (−0.084 ± 1.194)
Relative peak power (W/kg)	46.44 ± 3.62 (−0.366 ± 0.797)	50.55 ± 2.27 (0.291 ± 0.829)	49.36 ± 3.15 (−0.201 ± 0.930)	46.30 ± 3.56 (−0.356 ± 0.567)
RSI (m/m·s)	1.2 ± 0.3 (−0.256 ± 0.988)	1.7 ± 0.4 (0.383 ± 0.846)	1.8 ± 0.4 (0.219 ± 0.863)	1.2 ± 0.4 (−0.202 ± 0.845)
20 m sprint (s)	3.11 ± 0.19 (0.233 ± 0.949)	2.98 ± 0.13 (−0.333 ± 0.942)	2.97 ± 0.09 (−0.328 ± 0.537)	3.18 ± 0.19 (0.312 ± 0.906)
20 m momentum (m·s^−1^)	635.4 ± 76.7 (0.517 ± 0.745)	574.5 ± 57.1 (−0.792 ± 0.709)	647.6 ± 85.5 (−0.036 ± 0.952)	592.8 ± 68.8 (0.075 ± 1.176)
*V*O_2_max (mL∙kg^−1^·min^−1^)	47.6 ± 5.0 (−0.333 ± 0.973)	52.7 ± 3.1 (0.517 ± 0.748)	54.2 ± 5.1 (0.299 ± 1.006)	46.7 ± 2.9 (−0.323 ± 0.844)
*Psychological*				
Factor 1—adverse response to failure (AU)	2.7 ± 0.6 (−0.083 ± 0.691)	3.2 ± 0.9 (0.117 ± 1.321)	3.1 ± 0.7 (−0.099 ± 1.003)	2.7 ± 0.7 (−0.285 ± 0.822)
Factor 2—imagery and active preparation (AU)	3.8 ± 0.8 (0.006 ± 0.978)	3.7 ± 0.9 (1.619 ± 0.990)	3.6 ± 0.7 (−0.262 ± 0.757)	3.7 ± 1.1 (0.050 ± 1.194)
Factor 3—self-directed control and management (AU)	4.4 ± 0.6 (−0.239 ± 0.996)	4.7 ± 0.5 (0.350 ± 0.847)	4.5 ± 0.8 (0.242 ± 0.971)	4.6 ± 0.6 (0.162 ± 1.072)
Factor 4—perfectionistic tendencies (AU)	3.1 ± 0.6 (−0.078 ± 0.869)	3.3 ± 0.6 (0.117 ± 1.128)	3.3 ± 0.4 (−0.025 ± 0.999)	3.0 ± 0.8 (−0.379 ± 0.997)
Factor 5—seeking and using social support (AU)	4.6 ± 0.6 (0.117 ± 0.875)	4.4 ± 0.7 (−0.158 ± 1.108)	4.5 ± 0.6 (−0.014 ± 0.938)	4.6 ± 0.6 (0.116 ± 0.836)
Factor 6—active coping (AU)	4.4 ± 0.5 (−0.289 ± 0.896)	4.7 ± 0.6 (0.442 ± 0.931)	4.3 ± 0.5 (−0.099 ± 0.962)	4.6 ± 0.6 (0.299 ± 1.005)
Factor 7—clinical indicators (AU)	2.0 ± 0.5 (0.167 ± 0.999)	2.0 ± 0.5 (−0.242 ± 0.866)	2.2 ± 0.4 (−0.012 ± 0.742)	1.8 ± 0.5 (−0.252 ± 0.987)
*Perceptual-cognitive expertise*				
PCE (AU)	3.1 ± 1.3 (0.033 ± 0.970)	2.8 ± 1.5 (−0.067 ± 1.013)	2.3 ± 1.5 (−0.431 ± 0.960)	2.7 ± 1.1 (−0.188 ± 0.907)
*Social identity*				
In group ties (AU)	6.1 ± 1.0 (0.128 ± 0.883)	5.7 ± 1.1 (−0.208 ± 1.052)	5.9 ± 0.8 (−0.047 ± 0.912)	5.8 ± 1.4 (−0.117 ± 1.212)
Cognitive centrality (AU)	4.9 ± 1.5 (0.083 ± 0.978)	5.2 ± 1.5 (−0.100 ± 0.989)	5.3 ± 1.7 (0.004 ± 1.034)	5.3 ± 1.6 (0.045 ± 1.160)
In group affect (AU)	6.6 ± 0.7 (0.061 ± 1.035)	6.5 ± 0.5 (−0.050 ± 0.923)	6.6 ± 0.5 (0.178 ± 0.871)	6.7 ± 0.4 (0.222 ± 0.818)
Total score SIQ (AU)	5.8 ± 0.9 (0.094 ± 0.967)	5.8 ± 0.9 (−0.158 ± 1.000)	5.9 ± 1.0 (0.030 ± 0.959)	5.9 ± 0.9 (0.037 ± 1.112)

Note: Shows descriptive difference between forwards and backs and top-10 and bottom-10 potentials. BQs = birth quartiles; IMD decile = index of multiple deprivation decile; SIQ = social identity questionnaire; IHE = isometric hip extension; CMJ = countermovement jump; RSI = reactive strength index; PCE = perceptual-cognitive expertise; *V*O_2_max = maximal aerobic capacity; AU = arbitrary unit.

**Table 2 sports-10-00013-t002:** MANOVA results for socioeconomic, social identity, anthropometric, physical, psychological, and sport activity factors, as well as ANOVA results for perceptual-cognitive expertise and participation history.

Factor	All Forwards vs. Backs		Top-10 vs. Bottom-10 Potentials	
	*p*	F	*p*	F
**Socioeconomic**	0.030 *	3.985	0.049 *	3.581
**Social identity**	0.918	0.231	0.963	0.144
**Anthropometric**	<0.001 **	22.135	0.604	0.519
**Physical**	0.004 *	4.340	0.784	0.548
**Psychological**	0.273	1.354	0.954	0.273
**Perceptual-cognitive expertise**	0.788	0.074	0.550	0.371
**Sport activities**	0.172	1.678	0.018 *	3.820
**Participation history**	0.270	1.268	0.963	0.002

Note: Significance set for *p* = 0.05; * denotes a statistical significance of ≤0.05; ** denotes a statistical significance of ≤0.001.

**Table 3 sports-10-00013-t003:** Welch’s *t*-tests for forwards and backs and top-10 and bottom-10 potentials.

Characteristic	Welch’s *t*-Test	Cohen’s *d*
	(*p*)	
**Number of sports**		
Forwards vs. backs	0.275	−0.41 (−1.15; 0.32)
Top-10 potentials vs. bottom-10 potentials	0.963	−0.21 (−0.87; 0.83)
**Game exposure U8-U11**		
Forwards vs. backs	0.187	−0.51 (−1.25; 0.24)
Ranked top 10 vs. ranked bottom 10	0.003 *	0.80 (0.52; 2.48)
**Coach-led U8-U11**		
Forwards vs. backs	0.153	0.53 (−0.21; 1.26)
Top-10 potentials vs. bottom-10 potentials	0.266	0.50 (−0.37; 1.36)
**Peer-led U8-U11**		
Forwards vs. backs	0.296	0.37 (−0.37; 1.10)
Top-10 potentials vs. bottom-10 potentials	0.404	0.37 (−0.49; 1.23)
**Game exposure U12-U15**		
Forwards vs. backs	0.922	0.03 (−0.69; 0.76)
Top-10 potentials vs. bottom-10 potentials	0.661	0.19 (−0.66; 1.05)
**Coach-led U12-U15**		
Forwards vs. backs	0.439	0.29 (−0.44; 1.02)
Top-10 potentials vs. bottom-10 potentials	0.802	0.11 (−0.74; 0.96)
**Peer-led U12-U15**		
Forwards vs. backs	0.765	−0.11 (−0.84; 0.61)
Top-10 potentials vs. bottom-10 potentials	0.038 *	0.97 (0.05; 1.87)
**Town population**		
Forwards vs. backs	0.177	0.55 (−0.20; 1.30)
Top-10 potentials vs. bottom-10 potentials	0.880	−0.06 (−0.92; 0.79)
**IMD decile**		
Forwards vs. backs	0.020 *	0.88 (0.11; 1.64)
Top-10 potentials vs. bottom-10 potentials	0.015 *	−0.79 (−1.08; −0.22)
**In group ties**		
Forwards vs. backs	0.372	0.34 (−0.39; 1.08)
Top-10 potentials vs. bottom-10 potentials	0.870	0.07 (−0.78; 0.92)
**Cognitive centrality**		
Forwards vs. backs	0.622	0.18 (−0.54; 0.91)
Top-10 potentials vs. bottom-10 potentials	0.913	−0.04 (−0.90; 0.80)
**In group affect**		
Forwards vs. backs	0.761	0.11 (−0.61; 0.84)
Top-10 potentials vs. bottom-10 potentials	0.905	−0.05 (−0.90; 0.80)
**Total score SIQ**		
Forwards vs. backs	0.499	0.25 (−0.48; 0.98)
Top-10 potentials vs. bottom-10 potentials	0.939	−0.03 (−0.89; 0.82)
**Body mass**		
Forwards vs. backs	<0.001 **	0.82 (0.53; 1.53)
Top-10 potentials vs. bottom-10 potentials	0.427	−0.35 (−1.21; 0.51)
**Height**		
Forwards vs. backs	0.890	−0.05 (−0.78; 0.68)
Top-10 potentials vs. bottom-10 potentials	0.504	0.29 (−0.56; 1.15)
**Handgrip**		
Forwards vs. backs	0.899	0.04 (−0.68; 0.77)
Top-10 potentials vs. bottom-10 potentials	0.505	0.29 (−0.56; 1.15)
**IHE**		
Forwards vs. backs	0.013 *	0.81 (0.25; 1.88)
Top-10 potentials vs. bottom-10 potentials	0.741	−0.14 (−1.00; 0.71)
**CMJ**		
Forwards vs. backs	0.050 *	−0.73 (−1.48; 0.02)
Top-10 potentials vs. bottom-10 potentials	0.959	−0.02 (−0.87; 0.83)
**Peak power**		
Forwards vs. backs	<0.001 **	0.81 (0.73; 0.90)
Top-10 potentials vs. bottom-10 potentials	0.718	−0.16 (−1.01; 0.70)
**Relative peak power**		
Forwards vs. backs	0.041 *	−0.80 (−1.56; −0.03)
Top-10 potentials vs. bottom-10 potentials	0.633	0.21 (−0.64; 1.07)
**RSI**		
Forwards vs. backs	0.070	−0.69 (−1.44; 0.06)
Top-10 potentials vs. bottom-10 potentials	0.280	0.48 (−0.39; 1.35)
**20 m sprint**		
Forwards vs. backs	0.121	0.59 (−0.15; 1.34)
Top-10 potentials vs. bottom-10 potentials	0.049 *	−0.83 (−1.72; 0.06)
**20 m momentum**		
Forwards vs. backs	<0.001 **	0.89 (0.90; 2.67)
Top-10 potentials vs. bottom-10 potentials	0.827	−0.09 (−0.95; 0.76)
** *V* ** **O_2_max**		
Forwards vs. backs	0.012 *	−0.98 (−1.74; −0.19)
Top-10 potentials vs. bottom-10 potentials	0.128	0.69 (−0.19; 1.57)
**Factor 1**		
Forwards vs. backs	0.637	−0.19 (−0.92; 0.54)
Top-10 potentials vs. bottom-10 potentials	0.624	0.21 (−0.64; 1.07)
**Factor 2**		
Forwards vs. backs	0.988	0.00 (−0.72; 0.73)
Top-10 potentials vs. bottom-10 potentials	0.459	−0.33 (−1.18; 0.53)
**Factor 3**		
Forwards vs. backs	0.094	−0.63 (−1.38; 0.11)
Top-10 potentials vs. bottom-10 potentials	0.851	0.08 (−0.77; 0.93)
**Factor 4**		
Forwards vs. backs	0.619	−0.19 (−0.92; 0.54)
Top-10 potentials vs. bottom-10 potentials	0.417	0.36 (−0.50; 1.22)
**Factor 5**		
Forwards vs. backs	0.478	0.27 (−0.46; 1.00)
Top-10 potentials vs. bottom-10 potentials	0.744	−0.14 (−1.00; 0.71)
**Factor 6**		
Forwards vs. backs	0.043 *	−0.80 (−1.55; 0.02)
Top-10 potentials vs. bottom-10 potentials	0.351	−0.41 (−1.27; 0.45)
**Factor 7**		
Forwards vs. backs	0.245	0.43 (−0.30; 1.17)
Top-10 potentials vs. bottom-10 potentials	0.544	0.27 (−0.59; 1.12)
**PCE**		
Forwards vs. backs	0.790	0.86 (0.10; −0.63)
Top-10 potentials vs. bottom-10 potentials	0.550	−0.26 (−1.12; 0.59)

Note. Shows difference between forwards and backs and top-10 and bottom-10 potentials post-hoc and Cohen’s d effect size (90% confidence interval). IMD decile = index of multiple deprivation decile; SIQ = social identity questionnaire; IHE = isometric hip extension; CMJ = countermovement jump; RSI = reactive strength index; PCE = perceptual-cognitive expertise; *V*O_2_max = maximal aerobic capacity; * denotes a statistical significance of ≤0.05; ** denotes a statistical significance of ≤0.001.

## Data Availability

MDPI Research Data Policies.

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
