# Peer review of "A Multidisciplinary Investigation into the Talent Development Processes in an English Premiership Rugby Union Academy: A Preliminary Study through an Ecological Lens"

_sports, 2022, doi:10.3390/sports10020013_

Round 1

Reviewer 1 Report

INTRODUCTION:

Page 2, line 70-73 -“Indeed, questions remain surrounding the most appropriate processes that facilitate players’ progression towards RU senior professional status, since sport organisations’ modus operandi can often result in missing future professionals (Till, Weakley et al., 2020)” – This sentence in important but is not fully clear. Please provide more detailed explanation.

Please provide more details about position differences in rugby. Readers that are not so familiar with rugby should get some info about their in-game roles and possible differences obtained in previous studies.

METHOD:

Page 2, line 87-89 – “They were also ranked on their potential to become a senior professional RU player, regardless of playing position and age, from one to thirty by three Level 4 academy coaches.” – What does level 4 academy coach means? What exact criteria they used for classification?

Page 3, line 103-105 – “Psychological, socioeconomic, social identity, and participation history were collected using online questionnaires (Online surveys Jisc, Bristol, UK) that participants were asked to complete in their own time.” – What is the name of the questionnaire? Was it validated before? Please add references.

Page 3, Line 100-101 – “Each anthropometric and physical test was explained and demonstrated with each assessment preceded by a standardised warm-up.” – How the warm-up protocol looked like?

Page 4, line 152-155 – “After circa ninety seconds of recovery, players then completed three attempts for the RSI test whereby they performed ten consecutive jumps trying to reach maximal height for every bounce whilst spending as little time in contact with the ground between jumps as possible.” – RSI stands for reactive strength index. Please add this.

RESULTS:

No objections!

DISCUSSION:

Page 9, line 272-274 – “The aim of this study was to examine a range of factors that characterised playing positions and player ranks in an English Premiership RU academy through an ecological dynamics lens” – There is no need to repeat study aim as you stated it in the last chapter of introduction.

Page 9-10, line 283-286 – “These findings offer a preliminary study to better understand the TD processes in RU, provide professional RU academies a novel approach of assessing players, and establish a methodological framework that may be useful for other researchers in the future.” – This sentence is more suitable for last part of discussion.

Page 10, line 287-288 – “The IMD decile indicated that forwards originate from less deprived areas compared to backs.” – What is the reason for this finding? What this implies? I found it hard to realize it from this paragraph.

Page 10, 335-336 – “Therefore, coaches should consider the importance of body mass in developing players and their progression across an academy”- What about biological age? Coaches should be aware that players of some chronological age can differ a lot in their maturity status and therefore should be especially careful when selecting players based on morphology status.

Page 13, Limitations and Future directions – Considering that sample in this study consisted of players from 16 to 21 years, this should be highlighted as a major limitation as no age related differences were investigated and there is no data about age distribution between positions and rank categories (i.e. it is possible that age influenced this categorization)

CONCLUSION:

 No objections!

SUGESSTIONS:

  • References should be numbered according to journal’s instructions. Please amend it accordingly.
  • Topic is interesting and a huge number of variables is certainly strength of the studies, but this is preliminary study with several limitations and therefore can be improved in future.

Author Response

Responses to reviewer for the manuscript: “A Multidisciplinary Investigation into the Talent Development Processes in an English Premiership Rugby Union Academy: A Preliminary Study through an Ecological Lens”

INTRODUCTION:
Page 2, line 70-73 -“Indeed, questions remain surrounding the most appropriate processes that facilitate players’ progression towards RU senior professional status, since sport organisations’ modus operandi can often result in missing future professionals (Till, Weakley et al., 2020)” – This sentence in important but is not fully clear. Please provide more detailed explanation.
Thank you for the feedback. This has now been addressed in the manuscript.
Please provide more details about position differences in rugby. Readers that are not so familiar with rugby should get some info about their in-game roles and possible differences obtained in previous studies.
Thank you for the comment. We have now added this information with relevant references.
METHOD:
Page 2, line 87-89 – “They were also ranked on their potential to become a senior professional RU player, regardless of playing position and age, from one to thirty by three Level 4 academy coaches.” – What does level 4 academy coach means? What exact criteria they used for classification?
Level 4 coaching, also known as the England Rugby Performance Coaching Award, is the highest level of coaching qualification in England for Rugby Union and it is achieved after years of practice and experience. Coaches classified players according to their subjective vision of the game and in agreement to their coaching philosophy. This aspect has now been added.
Page 3, line 103-105 – “Psychological, socioeconomic, social identity, and participation history were collected using online questionnaires (Online surveys Jisc, Bristol, UK) that participants were asked to complete in their own time.” – What is the name of the questionnaire? Was it validated before? Please add references.
Thank you for the feedback. After a meeting, this group of researchers confirmed that the questionnaires mentioned by the reviewer were described and referenced individually in the paragraphs below that sentence. However, authors felt that was appropriate to add explanations in regards of the validation of such tests, therefore amendments have been now reported in the manuscript.

Page 3, Line 100-101 – “Each anthropometric and physical test was explained and demonstrated with each assessment preceded by a standardised warm-up.” – How the warm-up protocol looked like?
The warm up followed the Raise, Activate, Mobilise, and Potentiate (RAMP) protocol which allowed athletes to RAISE their metabolic indicators (heart rate and breathing), ACTIVATE muscles, MOBILISE joints, POTENTIATE movement (Jeffreys, I. (2019). The warm-up: Maximize performance and improve long-term athletic development. Human Kinetics). This protocol is normally used by different sport clubs and have been already used in previous literature (e.g. Dimundo, F., Cole, M., Blagrove, R. C., McAuley, A. B. T., Till, K., Hall, M., Pacini, D., & Kelly, A. L. (2021). The anthropometric, physical, and relative age characteristics of an English Premiership rugby union academy. International Journal of Strength and Conditioning, 1(1). https://doi.org/10.47206/ijsc.v1i1.67).
Page 4, line 152-155 – “After circa ninety seconds of recovery, players then completed three attempts for the RSI test whereby they performed ten consecutive jumps trying to reach maximal height for every bounce whilst spending as little time in contact with the ground between jumps as possible.” – RSI stands for reactive strength index. Please add this.
This has now been added. Thank you for the heads up.
RESULTS:
No objections!
This group is pleased that results have been found appropriate by the reviewer.
DISCUSSION:
Page 9, line 272-274 – “The aim of this study was to examine a range of factors that characterized playing positions and player ranks in an English Premiership RU academy through an ecological dynamics lens” – There is no need to repeat study aim as you stated it in the last chapter of introduction.
Thank you for the observation. This has now been deleted.
Page 9-10, line 283-286 – “These findings offer a preliminary study to better understand the TD processes in RU, provide professional RU academies a novel approach of assessing players, and establish a methodological framework that may be useful for other researchers in the future.” – This sentence is more suitable for last part of discussion.
After a careful consideration, authors agreed on this comment. Please see this sentence moved at the end of the discussion paragraph.

Page 10, line 287-288 – “The IMD decile indicated that forwards originate from less deprived areas compared to backs.” – What is the reason for this finding? What this implies? I found it hard to realize it from this paragraph.
These issues have been now addressed in the study. Thank you.
Page 10, 335-336 – “Therefore, coaches should consider the importance of body mass in developing players and their progression across an academy”- What about biological age?
Coaches should be aware that players of some chronological age can differ a lot in their maturity status and therefore should be especially careful when selecting players based on morphology status.
This has now been added into the text.
Page 13, Limitations and Future directions – Considering that sample in this study consisted of players from 16 to 21 years, this should be highlighted as a major limitation as no age related differences were investigated and there is no data about age distribution between positions and rank categories (i.e. it is possible that age influenced this categorization).
This has now been added in limitations and future directions section.
CONCLUSION:
No objections!
Authors are pleased that conclusions have been found appropriate by the reviewer.
SUGESSTIONS:
References should be numbered according to journal’s instructions. Please amend it accordingly.
This has now been amended.
Topic is interesting and a huge number of variables is certainly strength of the studies, but this is preliminary study with several limitations and therefore can be improved in future.
Authors thanks the reviewer for this comment. The research group appreciate that this paper could be valuable and could provide an amount of reliable variables useful for the sport science network. Authors provided amendments according to the reviewer suggestions and hope that the present
paper could be used for future reference.

Reviewer 2 Report

1) I apologize for the delay

Overall, the present study is solid and provides valuable information for coaches. I would ask that the authors provide some more detail as it relates to their methods. 

  • please describe how the ranking system works (this may be common knowledge for those in the geographical location of the authors,) - this is particularly important, and believe it would be best to err on the side of more detail 
  • please add substantial detail for the strength assessment, was there a familiarization period?  please describe the warm up, time of day, number of trials, explain how the position was controlled, provide specs of the platform and more information of dynamometer used. 
  • please add similar info for the grip test 
  • were all tests performed on the same day? 
  • for jumps was arm swing included? 
  • why did you choose to use best vs avg, for non fatiguing tests avg is recommended (see Sands and Stone "Are you progressing and how would you know" from Olympic Coach) 
  • similar questions for sprint test - more info needed
  •  

Author Response

1) I apologize for the delay
Authors understand the critical pandemic time that we are all living and accept the apologises.
These are not needed anyway.

Overall, the present study is solid and provides valuable information for coaches. I would ask that
the authors provide some more detail as it relates to their methods.
This group thanks the reviewer for the appreciation of the present manuscript. Authors are happy
that their contribution can be of any valuable use for practitioners and provided more information
as requested.
Please describe how the ranking system works (this may be common knowledge for those in the
geographical location of the authors,) - this is particularly important, and believe it would be best
to err on the side of more detail.
This has now been added. As responded to the previous comment of the revisor one, coaches used
a very subjective ranking system according both to their philosophy of coaching and vision of the
game. The evaluation was based on their experience as ex-players and navigated coaches.
Please add substantial detail for the strength assessment, was there a familiarization period? please
describe the warm up, time of day, number of trials, explain how the position was controlled,
provide specs of the platform and more information of dynamometer used.
Thank you for the comment. No electronics platforms were used. Please see this information
added in the text.
Please add similar info for the grip test
Please see section amended.
Were all tests performed on the same day?
Yes, tests were performed on the same day.
For jumps was arm swing included?
No. During jumps, arms had to be placed on the hips.
Why did you choose to use best vs avg, for non fatiguing tests avg is recommended (see Sands and
Stone "Are you progressing and how would you know" from Olympic Coach)
Authors thanks the reviewer for the interesting comment. Although this group is aware of the
potential benefits of using average procedure, authors followed adapted tests protocols as reported
in Fukuda (2018) [Fukuda, D. H. (2018). Assessments for sport and athletic performance. Human
Kinetics].
Similar questions for sprint test - more info needed.
Please see information added in the text.

Round 2

Reviewer 1 Report

Thank You for amending your manuscript according my suggestions.

Congratulations on great work!